# Measuring frailty in younger populations: a rapid review of evidence

Gemma F Spiers [1], Tafadzwa Patience Kunonga,[1] Alex Hall,[2] Fiona Beyer,[1] Elisabeth Boulton [2], Stuart Parker,[1] Peter Bower,[2] Dawn Craig,[1] Chris Todd,[2] Barbara Hanratty [1]

[1]National Institute for Health Research (NIHR) Older People and Frailty Policy Research Unit, Population Health Sciences Institute, Newcastle University, Newcastle upon Tyne, UK
[2]National Institute for Health Research (NIHR) Older People and Frailty Policy Research Unit, School of Health Sciences, Faculty of Biology, Medicine and Health, University of Manchester, Manchester, UK

**Correspondence to**
Dr Gemma F Spiers;
gemma-frances.spiers@newcastle.ac.uk

## ABSTRACT

**Objectives** Frailty is typically assessed in older populations. Identifying frailty in adults aged under 60 years may also have value, if it supports the delivery of timely care. We sought to identify how frailty is measured in younger populations, including evidence of the impact on patient outcomes and care.

**Design** A rapid review of primary studies was conducted.

**Data sources** Four databases, three sources of grey literature and reference lists of systematic reviews were searched in March 2020.

**Eligibility criteria** Eligible studies measured frailty in populations aged under 60 years using experimental or observational designs, published after 2000 in English.

**Data extraction and synthesis** Records were screened against review criteria. Study data were extracted with 20% of records checked for accuracy by a second researcher. Data were synthesised using a narrative approach.

**Results** We identified 268 studies that measured frailty in samples that included people aged under 60 years. Of these, 85 studies reported evidence about measure validity. No measures were identified that were designed and validated to identify frailty *exclusively* in younger groups. However, in populations that included people aged over *and* under 60 years, cumulative deficit frailty indices, phenotype measures, the FRAIL Scale, the Liver Frailty Index and the Short Physical Performance Battery all demonstrated predictive validity for mortality and/or hospital admission. Evidence of criterion validity was rare. The extent to which measures possess validity across the younger adult age (18–59 years) spectrum was unclear. There was no evidence about the impact of measuring frailty in younger populations on patient outcomes and care.

**Conclusions** Limited evidence suggests that frailty measures have predictive validity in younger populations. Further research is needed to clarify the validity of measures across the adult age spectrum, and explore the utility of measuring frailty in younger groups.

## BACKGROUND

Frailty is characterised by increased vulnerability to stressors, and has been described as a problematic expression of population ageing.[1 2] The presence of frailty is associated with a number of adverse outcomes, including lower quality of life and an increased risk of functional decline, admission to hospital or long-term care and mortality.[3–6] Recent estimates place the global incidence of frailty at 43.4 cases per 1000 person-years, although there is substantial variation by country-level income and disease-specific populations.[7]

Debates about how to measure and operationalise frailty have led to two dominant models: the phenotype model and the cumulative deficit model. The phenotype model, which was conceptualised through clinical observation and epidemiological study, operationalises frailty as the presence of three or more of the following criteria: exhaustion, weight loss, weakness/loss of muscular strength, reduced gait speed and reduced energy/physical activity.[8] In contrast, the cumulative deficit model, which was developed through consideration of biological theories of ageing,[9] considers frailty an accumulation of deficits including clinical signs and symptoms, diseases and disability. In this model, 'the more things somebody has wrong with them, the more likely they are to be frail'[10] (p.722).

There are numerous tools to measure frailty, which are not necessarily interchangeable.[11] A recent review of measures in older populations identified 51 separate instruments.[12] Although many measures of frailty are informed by the cumulative deficit or

phenotype models, there is still much variation in the content of these instruments.[11] In one analysis of 35 frailty measures, there were varying levels of agreement for identifying who was frail between tools.[13] There are further challenges in assessing the risk of frailty in clinical settings, where different approaches exist across primary, secondary and specialist care.[2] For example, the Clinical Frailty Scale, gait speed, the seven-item Program of Research on the Integration of Services for the Maintenance of Autonomy and timed-up-and-go tests are all validated tools to assess the risk of frailty in older populations in clinical care.[14–16] These numerous instruments present challenges to researchers and clinicians when choosing how to conceptualise, measure and assess frailty.[12]

► Comprehensive and systematic searches captured the core evidence about measuring frailty in younger populations.

► We were only able to judge predictive validity based on frailty outcomes for older populations and not younger populations.

► The focus on younger groups offers a novel addition to our understanding of frailty measurement.

► Our exclusion of studies with samples aged, on average, over 60 years means we have omitted some studies that include younger participants.

The prevalence of frailty increases with age and so it is typically assessed in older populations. In 2019, the International Conference of Frailty and Sarcopenia Research Task Force recommended routine frailty screening in populations aged over 65 years.[17] This reflects current clinical guidance in the UK, for example, where screening for frailty in populations aged over 65 years is routine in primary care, and also forms part of the Comprehensive Geriatric Assessment in secondary care settings.[15] Such routine screening is part of efforts to support people as they age.

The utility of identifying frailty in adults aged under 60 years also warrants scrutiny. Some vulnerable groups with chronic conditions may be at risk of becoming frail earlier in life, while the onset of illness and disability occurs at younger ages for people living in areas of greater deprivation.[18–20] There is, therefore, potential for younger groups to be classed as frail, and identification may aid the delivery of timely care. Furthermore, studies are increasingly measuring frailty prevalence across the life span.[21–23] Yet, current debates and evidence about how to measure frailty remain focused on older populations. A number of systematic reviews have examined the validity of various frailty measures, most recently in 2019.[12 24 25] These reviews focus on the measurement of frailty in older populations, and there remains a lack of clarity about the validity of frailty measures in younger groups.

To address this gap, we aimed to synthesise evidence on how frailty is measured in younger populations (aged <60 years). Specifically, we sought evidence to address the following questions:

1. Are there any validated tools to identify frailty specifically in younger populations?
2. How are existing tools to identify frailty in older populations used to measure and identify frailty in younger age groups, and are such tools validated in these younger populations?
3. How is frailty identified in younger people with long-term or life-limiting conditions?
4. In what other ways (other than a validated tool) is frailty in younger populations currently operationalised and identified from existing data in observational and experimental studies?

5. Is there any evidence of impact (on patient outcomes, staff, clinical management or health service utilisation) of identifying frailty in younger populations, including identification of opportunities for intervention?

## METHODS

We conducted a rapid review of primary studies. Rapid reviews offer a streamlined version of standard systematic review methods in a shorter time frame, with the resultant output a summary, rather than in-depth synthesis, of evidence.[26] There is no agreed methodology for the conduct of rapid reviews, but for the purposes of this work, rapid methods included: systematic searches, double screening of titles and abstracts for 5% of studies to pilot the eligibility criteria; single researcher screening of full texts; second checking of data extraction for 20% of studies and omitting an assessment of study quality, an approach that has been used in other reviews of frailty measures.[25]

### Search strategy

A search strategy was developed, piloted and tested, based on three concepts: frailty, measurement and age (see online supplemental material 1 for the strategy applied to MEDLINE). The search strategy excluded studies that were indexed with terms for older people, unless the record also contained indexing terms pertaining to a younger population ('middle aged' or 'young adult').

Searches were carried out in: MEDLINE (OVID) (1946–March week 1 2020); PsycINFO (OVID) (1967–March week 2 2020); CINAHL (EBSCO) (1982–March 2020); Science Citation Index (Web of Knowledge) (1970–March 2020); Social Science Citation (Web of Knowledge) (1970–March 2020). Three sources of grey literature were also searched: OpenGrey,[27] Social Care Online,[28] and the National Institute of Health and Care Excellence.[29] References of relevant systematic reviews were also checked. Searches were carried out in March 2020.

### Review criteria

Eligible for inclusion were primary research studies that measured frailty in populations aged under 60 years, and published after 2000 in English, using experimental or observational study designs (table 1). The age threshold of below 60 years was selected, as opposed to 65 years, as initial scoping indicated that studies on frailty in older populations typically used samples aged 60 years and over. Studies that included those aged 60 years and over were included only if the majority of sample was aged under 60 years. Where it was not possible to determine this majority, studies were included if the mean or median sample age was between 18 and 59.9 years. As the concept of frailty began to develop from around 2001,[30] we excluded studies before 2000 to prioritise the most relevant and contemporary evidence.

**Table 1** Review inclusion/exclusion criteria

|  | Include | Exclude |
|---|---|---|
| Population | Adults aged 18–59.9 years.<br>Studies that include those aged ≥60 years if the majority of sample is aged under 60 years. Where it is not possible to determine if the majority of the sample is aged under 60 years, studies where the mean/median sample age was under 60 years were included.<br>Any condition/diagnosis. | Studies of people aged ≥60 years only, or where the majority of the study sample is aged >60 years. |
| Exposure/outcome | Any measure of frailty that is validated, including those currently used in 65+ populations and those developed specifically for younger age groups.<br>Any approach to identifying frailty from existing data.<br>Frailty may be either the exposure (eg, if using it to predict other outcomes) or the outcome (eg, if estimating prevalence). |  |
| Study design | Observational or experimental designs.<br>Published in English after 2000. |  |

## Study selection

Title and abstract screening was piloted by three researchers (AH, TPK, GFS) independently on 5% of records. Decisions were compared and discussed to resolve inconsistencies before proceeding. The titles and abstracts of the remaining 95% of records were then screened by single researchers (AH, TPK, GFS). The full texts of selected records were retrieved and assessed for inclusion against the review criteria. For studies that reported details of a validated frailty measure, the referenced publication was also retrieved and assessed against the review criteria.

## Data extraction and synthesis

A data extraction form was developed and piloted. Summary study information was extracted: author and date; country; measure used; sample age (mean, median and range where reported) and long-term condition of sample (if applicable). Included studies were further scrutinised to identify those that reported evidence that indicated whether the measure was validated. To ascertain validity, we adapted criteria used in a previous review of frailty measures in older populations.[31] That is, the validity of a frailty measure was ascertained through evidence that:

1. The measure predicted mortality and/or hospitalisations (*predictive validity*).
2. The measure was associated with another validated measure of frailty (*criterion validity*).

Predictive validity indicates whether the measure can predict outcomes of relevance. Mortality and hospital admissions were chosen as outcomes for the test of predictive validity as these are commonly associated with frailty.[32–35] Criterion validity indicates whether the new measure is assessing what it intends to, by comparing it with a validated measure of the same concept.[31 36] This is achieved by assessing the association between the two measures (eg, correlation or regression), or the extent to which two measures produce the same results (agreement).

For studies that reported evidence about predictive and/or criterion validity, data were extracted about: the measure (eg, items, cut-off scores used); whether the evidence indicated validity, including prediction estimates; and any evidence about the impact of measuring frailty in younger populations on patient outcomes, staff, clinical management or health service utilisation (eg, if a measure was trialled or evaluated). Data were extracted by single researchers (AH, TPK, GFS), with 20% checked by a second for accuracy.

A narrative synthesis was used to summarise what measures of frailty were used and which measures demonstrated evidence of validity (in which populations). To facilitate this, each study that reported evidence about validity was allocated to a colour-coded framework (online supplemental table 1). Studies coded green demonstrated evidence of validity (a statistically significant prediction of mortality and/or hospital admissions, or authors described associations or agreement between measures as good or strong). Studies coded amber reported inconsistent or unclear evidence of validity, or authors described associations or agreement between measures as fair or modest. Studies coded red demonstrated no evidence of validity (the prediction of mortality and/or hospital admissions was not statistically significant, authors described associations or agreement between measures as weak or poor, or data were not reported in the publication to verify a claim of an association). Finally, we summarised evidence about any reported impact of measuring frailty in younger populations on patient outcomes and care.

## Patient and public involvement

This work was a rapid response to a request by the UK Department of Health and Social Care with a limited time frame available for completion. Involving patients effectively in such rapid responsive work can be challenging, and in this case we did not feel that we could effectively engage with patients in the time available. However, the work was undertaken in response to a policy imperative.

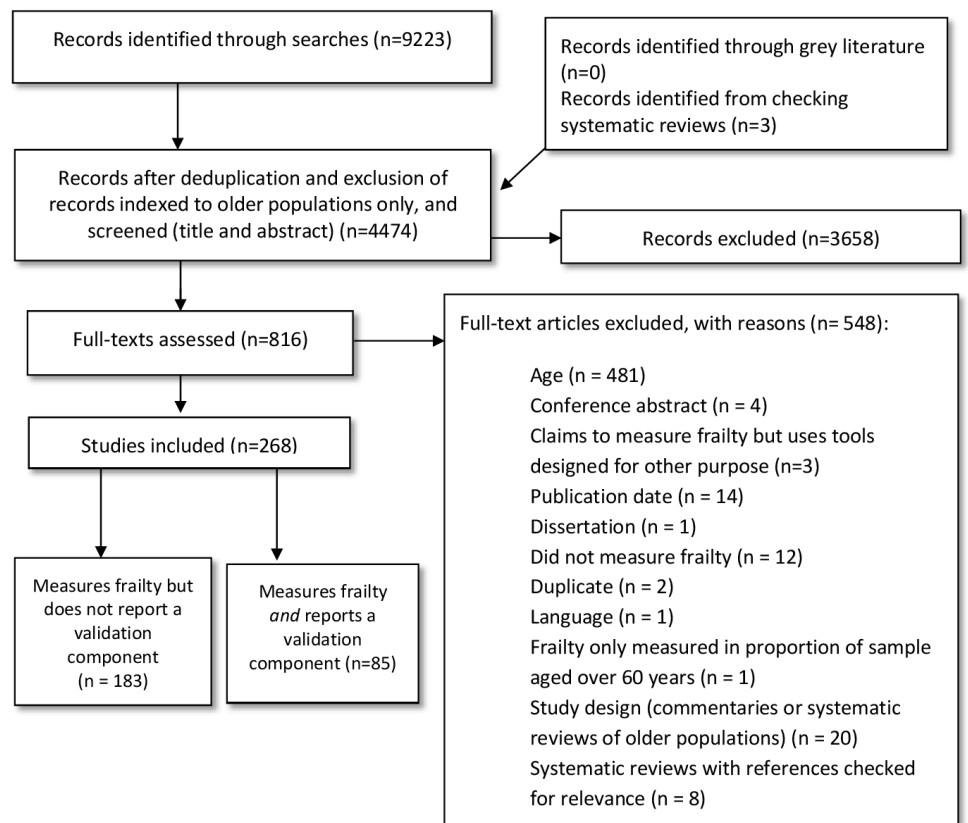

**Figure 1** Preferred Reporting Items for Systematic Reviews and Meta-Analyses flow chart.

## RESULTS

After screening, 268 publications met the review criteria (figure 1; full bibliographical details of the included studies are also listed in the online supplemental material 1).

Most studies used samples aged under *and* over 60 years. A minority of studies (n=6) used samples aged entirely below 60 years.[37–42] Just over 70% of studies used study samples with an average age between 51 and 59 years, while 24.1% used samples with an average age of between 41 and 50 years. Few studies measured frailty in samples with an average age below 40 years (figure 2). Thus, when referring to *younger populations* in these findings, this typically refers to the older end of the 18-59 year spectrum.

### Are there any validated tools to identify frailty specifically in younger populations?

No measures were identified that were designed and validated to identify frailty *exclusively* in younger (18–59.9 years) populations.

### How are existing tools to identify frailty in older populations used to measure and identify frailty in younger age groups, and are such tools validated in these younger populations?

Across studies, 41 measures of frailty were identified (online supplemental table 2). The most common were phenotype measures or cumulative deficit frailty indices. Some studies measured frailty using single, or a combination of, indicators, such as sarcopenia with vascular disease, or muscle mass. A small number of studies claimed to measure frailty but used tools not designed for this purpose (the Katz Index, the Activities of Daily Living Scale, the Karnofsky Performance Scale and the Braden Scale). The studies reporting these tools were therefore not included in this review.

A subset of 85 studies, of the 268 that met the review criteria, reported evidence about the predictive and/or criterion validity of their chosen frailty measure (online supplemental table 3).[21–23 43–124]

Across these studies, 13 measures were used (32 in total when accounting for different versions of cumulative deficit frailty indices). These measures included cumulative deficit frailty indices, phenotype measures, the FRAIL Scale, the Short Physical Performance Battery, the Clinical Frailty Scale, the Liver Frailty Index, the John Hopkins Frailty Indicator, and the Study of Osteoporotic Fracture Frailty Scale and Cardiovascular Health Study Frailty Scale.[21–23 43–94 96–124] Although termed a 'Liver Frailty Index', this measure did not comprise items that were specific to liver disease. Three measures were individual markers of frailty (muscle mass, walk test and hand grip), each used in isolation and not as part of a multiple item measure.[56 59 69 82] One measure was a surgical risk index, which incorporated a frailty assessment.[95]

Online supplemental table 3 summarises the evidence for each study. Evidence of criterion validity, where a measure was compared with a 'gold standard' validated frailty measure, was sparse. Most studies reported evidence about the association between the measure of

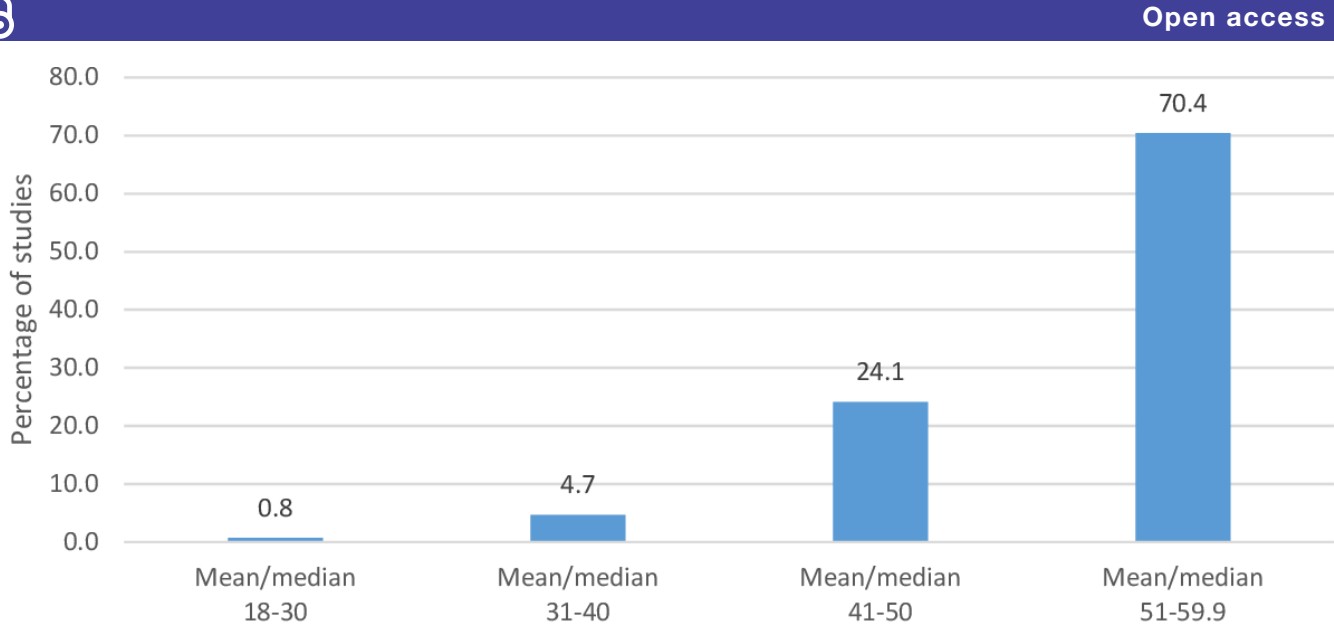

*ᵃCalculated from studies where a mean or median sample age was reported (n=257)*

**Figure 2** Proportion (%) of studies with different average sample ages.

frailty and mortality and/or hospital admissions, which we used to infer predictive validity.

There was evidence of predictive validity for cumulative deficit frailty indices and phenotype measures in general populations, diagnostic-specific populations (HIV, systemic lupus erythematosus, metabolic syndrome, survivors of myocardial infarction, childhood survivors of cancer, chronic kidney or end-stage renal disease) and surgical populations (haematopoietic cell transplant or other non-specified surgery).[22 23 43 46 51 52 55 58 62 65–67 72 76 77 80 81 83 88–93 99 101 105 110 112 114–116 118 121 123–125] For some populations, evidence of predictive validity was inconsistent for a cumulative deficit frailty index (intellectual disability)[94 105] and phenotype measures (heart failure and those undergoing lung or kidney transplant).[73 74 78 96–98 106 108 117 119 120]

There was also evidence of predictive validity for the FRAIL Scale (general population and population with diabetes),[53 54 64 93 109 116 122] the Liver Frailty Index (liver disease)[68 84 85 87] and the Short Physical Performance Battery (populations undergoing lung or kidney transplants and those with end-stage liver disease).[57 86 111 119 120]

For the remaining measures, there was either no evidence of predictive validity, or evidence was too inconsistent or uncertain to draw a conclusion.

In terms of whether the validity of a measure varied by age, only two studies stratified their analysis of prediction by age group. One study showed that a cumulative deficit frailty index based exclusively on laboratory test-based indicators of illness-predicted mortality only in groups aged 40–65 years and not those aged 20–39 years. In the same study, a cumulative deficit frailty index based entirely on self-reported items predicted mortality in both age groups, but prediction was statistically less certain for those aged 20–39 years.[21] In the second study, higher levels of frailty predicted mortality at all ages, but for the younger group (<39 years), differences in survival were only statistically significant between the least and most frail. We further explored the role of age by comparing evidence across studies according to the mean or median sample age (where this was reported). We used this approach for the two most commonly used measures: the frailty indices and the phenotype measures (figure 3A,B). With very few studies involving participants with an average age below 40 years, there was insufficient evidence to assess how the validity of these measures varied according to the average age of the sample in which the measure was tested.

### How is frailty identified in younger people with long-term or life-limiting conditions?

Most studies used frailty measures in a clinical subgroup and *not* a generic population (online supplemental table 3). Younger populations of people with life-limiting or long-term conditions, in which frailty measures demonstrated evidence of predictive validity, included: HIV (Frailty Index); chronic or end-stage kidney disease (Frailty Index, phenotype model, the Short Physical Performance Battery); diabetes (FRAIL Scale) and end-stage liver disease (Liver Frailty Index, the Short Physical Performance Battery).

### In what other ways (other than a validated tool) is frailty in younger populations currently operationalised and identified from existing data in observational and experimental studies?

Typically, studies operationalised and measured frailty using existing tools, the majority of which were cumulative

a

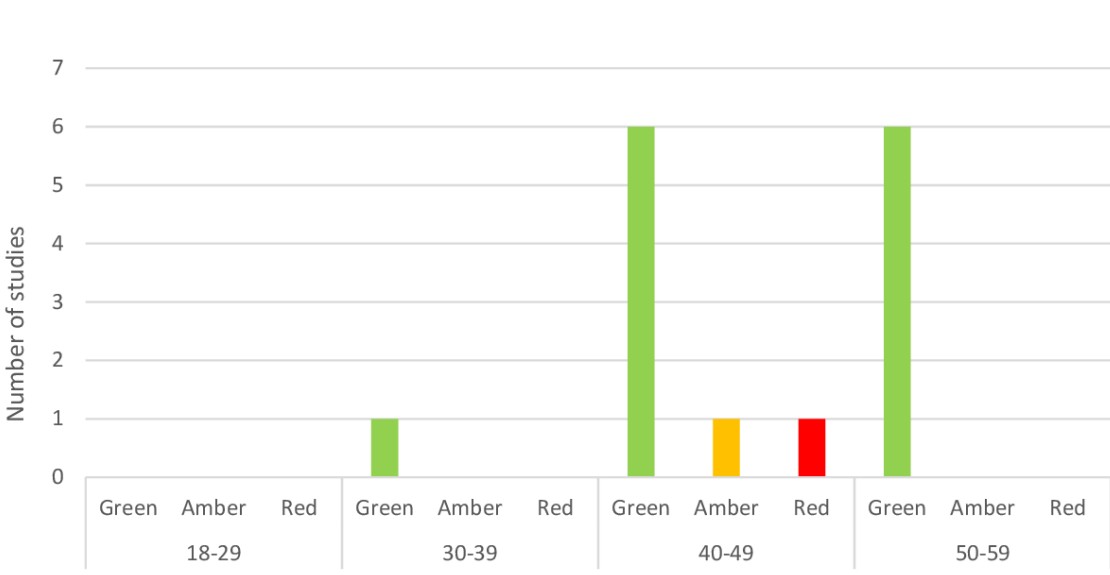

*Sample mean/median age (years)*
*Evidence of validity (green); mixed or uncertain evidence of validity (amber); evidence of no validity (red)*

The reader is referred to Table S3 for details of the studies using cumulative deficit frailty indices and their corresponding evidence classification.

b

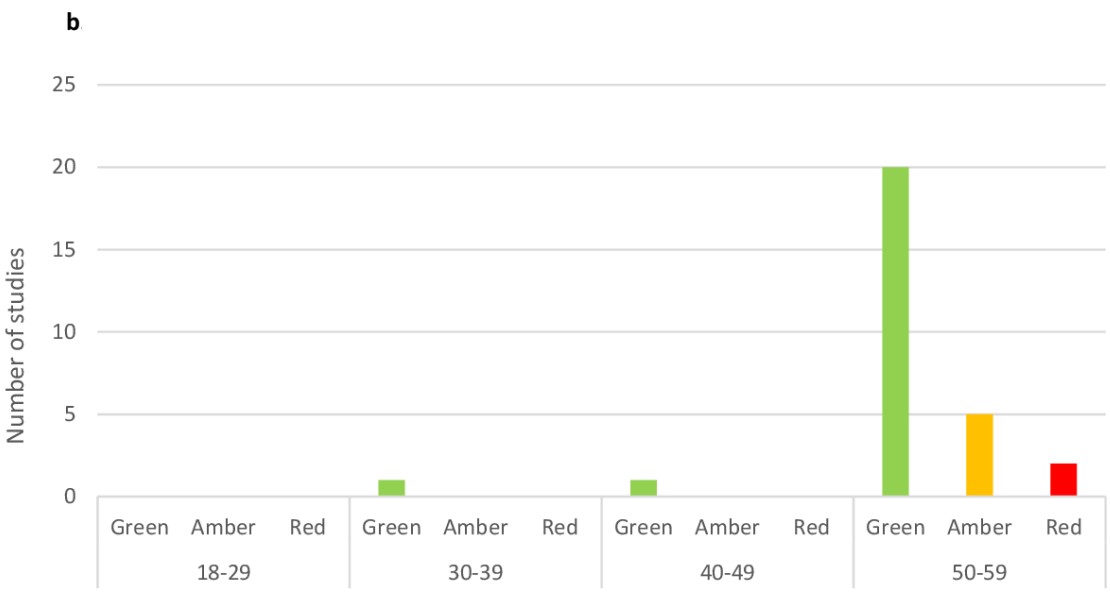

*Sample mean/median age (years)*
*Evidence of validity (green); mixed or uncertain evidence of validity (amber); evidence of no validity (red)*

The reader is referred to Table S3 for details of the studies using phenotype measures and their corresponding evidence classification.

**Figure 3** (A) Evidence of validity for cumulative deficit frailty indices by study sample mean/median age (years). (B) Evidence of validity for phenotype measures by study sample mean/median age (years).

deficit frailty indices or phenotype measures. In four studies, frailty was identified from single indicators (a walk test and grip strength, one study each, and muscle mass, two studies), which were not part of a multiple item measure.

### Is there any evidence of impact of measuring frailty in younger populations?

No studies were identified that evaluated the impact of measuring frailty in younger populations on patient outcomes, staff, clinical care or service utilisation.

## DISCUSSION

This review identified no measures that were designed and validated to measure frailty *only* in younger populations. However, in study populations that included people aged under 60 years, there was some evidence that cumulative deficit frailty indices, phenotype measures, the FRAIL Scale, the Short Physical Performance Battery and the Liver Frailty Index have predictive validity. These measures demonstrated validity in both general and single diagnosis populations, and those undergoing some types of surgical procedures. Evidence of predictive validity for other measures was either absent or less certain, and evidence of criterion validity was sparse.

Overall, the evidence suggests that some measures allow for the identification of frailty in younger populations. However, a more cautious interpretation should consider the role of sample age. Most studies used populations that included both people aged over and under 60 years of age. We also know that the prevalence of frailty is higher in older, rather than younger, groups.[126 127] Therefore, it is possible that any observed predictive validity was due to higher prevalence of frailty among older participants. Only two studies stratified their analysis of prediction by age group: both showed that the ability of the frailty measures to predict mortality was inconsistent between younger and older age groups.[21 23] We are also aware of another study that showed a phenotype measure does not predict mortality for women aged 37–45 years.[128] However, this study was not included in our review as it was not possible to determine if the majority of the sample was aged under 60 years. Furthermore, most study populations were aged, on average, over 40 years. Therefore, when concluding that measures observed validity in younger populations, we are unclear to what extent this reflects validity for the entire 18-59 year adult age spectrum.

We identified no studies that evaluated the impact of assessing frailty in younger populations. In many of the studies, there was speculation about the proposed benefits of assessing frailty, such as investigation of patient health status[66 109 122]; patient prognosis[54 61 64 70 129]; risk stratification and preoperative decision-making[60–63 65 68 84–86 92 98 101 108]; identifying early or preventive intervention[46 64 92 125] and patient counselling.[43 44 48 60 92] Despite these speculations, evidence is needed to clearly establish whether assessing frailty in younger groups does indeed make a difference

to patient care in the ways suggested. The absence of any evidence about the impact of measuring frailty is not specific to younger populations. Others have noted the need for evidence about the utility of frailty instruments in practice and whether such tools usefully inform clinical decision-making in care for older populations.[2]

A final observation about these findings concerns terminology. We identified a small number of studies where frailty was measured using tools designed for quantifying dependency and in one case, risk of pressure ulcers.[129–131] While dependency and frailty are related,[132–134] they are not conceptually equivalent, and the use of these tools was not clearly justified. The use of these measures signals either some conceptual ambiguity about the nature of frailty, or what Bouillon *et al*, in their 2013 review of frailty measures in older populations, call 'a lack of terminological rigour' (p.4).[25] The implications for this review were that we were unable to reliably interpret any evidence about the validity of these tools as measures of frailty, and were thus excluded.

### Strengths and limitations

The primary aim of this review was to identify how frailty was measured in younger populations, and the validity of such measures. A comprehensive search strategy using electronic databases, grey literature, systematic reviews and publications about measure validation referenced within included studies has captured the core literature to achieve this aim.

Our conclusions about validity are based on a synthesis of evidence that predominantly concerns the ability of measures to predict mortality and hospital admissions. We chose these outcomes to assess predictive validity as they are known consequences of frailty in older groups.[32–35] We extended this assumption to younger populations but we acknowledge the limitations of this: evidence about outcomes of frailty *specifically* in younger populations is far less established. We also judged predictive validity based on the statistical significance of prediction. This facilitated an efficient summary assessment of evidence for this rapid review. Future work could add further insight into the validity of measures by considering the size and strength of prediction, particularly if more studies that stratify analyses by age group become available.

We did not undertake a quality assessment due to the rapid nature of this review. While an assessment of study quality can identify the most robust studies, we do not believe such as assessment would change the conclusions of this review. A key finding is that we are uncertain to what extent the predictive validity of frailty measures in younger populations is driven by the presence of older sample participants, among whom frailty is more prevalent. Given this limitation of the evidence, identifying the most robust studies would not enhance our certainty about the validity of measures in younger groups.

Finally, our focus on frailty measurement in younger populations was operationalised by excluding studies where the majority were aged, or the average sample age

was, 60 years and over. We used the age threshold of 60, and not 65, years as initial scoping indicated that studies on frailty in older populations typically used samples aged 60 years and over. This criterion has inevitably excluded studies where frailty was measured in samples that were, on average, older than 60 years but nonetheless included younger groups. However, this criterion was necessary to focus on evidence about younger groups and to avoid duplicating existing reviews of frailty measurement in older populations. By using this approach, we have produced a novel review and addressed a notable gap in our understanding of frailty measurement.

## Implications for practice and future research

Identifying frailty in younger groups may have value if it aids the delivery of care for those who experience age-related illness and disability earlier in life. However, we identified no evidence about the impact of measuring frailty in younger groups, and the utility of doing so remains uncertain. Even if frailty assessment in healthcare is extended to those aged under 65 years, clearer evidence is needed about the validity of frailty measures *across* the younger adult age spectrum.

It is also worth noting that the measures identified in this review were no different to those used with older populations. This may reflect an assumption that frailty is homogeneous across older and younger groups, or it may simply be driven by the availability of frailty measures. Without evidence about the impact of measuring frailty in younger populations, it is impossible to judge whether this is appropriate. This also presents a question about construct validity: is frailty—a concept normally associated with ageing—the same, and thus measurable using existing tools, in younger populations? These are important issues to address and clarify should the assessment of frailty in younger populations be pursued in healthcare policy and clinical practice.

## Conclusions and implications

Approaches to measuring frailty in younger groups mirrored those used in older populations. Some of these measures of frailty demonstrated predictive validity in samples that *included* those aged under 60 years. However, further evidence is needed to draw clear conclusions about the validity of these measures across the adult age spectrum. Despite the potential utility of identifying frailty in younger populations, evidence about the impact on patient outcomes, staff, clinical management and health service utilisation is needed.

**Contributors** GFS co-designed the protocol, undertook all stages of the review and co-wrote the paper. TPK co-designed the protocol, undertook all stages of the review and co-wrote the paper. AH co-designed the protocol, undertook all stages of the review and co-wrote the paper. FB co-designed the protocol, undertook the searches for the review and co-wrote the paper. EB co-designed the protocol and co-wrote the paper. SP co-designed the protocol and co-wrote the paper. PB co-designed the protocol, contributed to the analysis of review data and co-wrote the paper. DC co-designed the protocol, contributed to the analysis of review data and co-wrote the paper. CT co-designed the protocol and co-wrote the paper. BH co-designed the protocol and co-wrote the paper.

**Funding** This report presents independent research funded by the National Institute for Health Research Policy Research Unit in Older People and Frailty (PR-PRU-1217-21502).

**Disclaimer** The views expressed are those of the authors and not necessarily those of the NIHR or the Department of Health and Social Care.

**Competing interests** None declared.

**Patient consent for publication** Not required.

**Provenance and peer review** Not commissioned; externally peer reviewed.

**Data availability statement** All data relevant to the study are included in the article or uploaded as supplemental information.

**ORCID iDs**
Gemma F Spiers http://orcid.org/0000-0003-2121-4529
Elisabeth Boulton http://orcid.org/0000-0003-2791-8295
Barbara Hanratty http://orcid.org/0000-0002-3122-7190

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
