## [Reviewer comments · BMJ Open]

ARTICLE DETAILS

TITLE (PROVISIONAL)	Measuring frailty in younger populations: a rapid review of evidence
AUTHORS	Spiers, Gemma; Kunonga, Tafadzwa; Hall, Alex; Beyer, Fiona; Boulton, Elisabeth; Parker, Stuart; Bower, Peter; Craig, Dawn; Todd, Chris; Hanratty, Barbara

VERSION 1 – REVIEW

REVIEWER	J. Gordon Boyd, MD, PhD, FRCPC Queen's University Canada
REVIEW RETURNED	02-Dec-2020

GENERAL COMMENTS	Thank you for the opportunity to review this manuscript by Spiers and colleagues. The authors have performed a rapid review of the literature from 2000 onwards with the objective of furthering our understanding the consequences of frailty for younger individuals (<60 years). As there are no standardized definitions of what constitutes a “rapid review”, they authors have clearly defined the parameters of their rapid review, which included a systematic search of 4 databases and 3 sources of grey literature, single researcher performing title/abstract screening (with 5% double screened), single researcher performing data abstraction (20% were duplicated for accuracy), and omitting the assessment of study quality. Two concepts were explored, predictive validity (the measure predicted mortality/hospitalizations) and criterion validity (the measure was associated with another measure of frailty). Over 9000 articles were identified, with ultimately 268 selected for full review. Eighty-five of those articles addressed either predictive or criterion validity. While this is a study of the role of frailty in younger individuals, the majority of the data included in this review is from individuals 50-59. In the 85 studies identified that assessed validity of frailty assessments in younger patients, 13 frailty assessment tools were used. Little data was available for criterion validity. For predictive validity, frailty assessment tools were associated with increased mortality and hospitalization across several cohorts, including HIV patients and childhood survivors of cancer.
--

	This is such an important topic. I thank the authors for taking this on, and for the tremendous amount of work that has gone into this study. All of the essential data is there, but I think the study could be strengthened by addressing the following concerns: Issues: Introduction: 5 questions-where are the 5 answers? Some of the questions line up with section headers in the results section, but not all of them. Was there a hypothesis for this study? Methods: Why was it not possible to involve the public? Engaging patients and and/or families might be incredibly instructive, especially when it comes to answering the 5th question (eg. what is the impact of frailty screening in younger patients). For clarity, I am not suggesting that engaging the public is a necessary revision for this manuscript, rather that it is insufficient to state that "it is not possible". One of the key questions of the study is whether or not frailty measures are valid for patients under the age of 60. However, the rating scale (green, amber, red) isn't mentioned in the methodology, but appears in the supplemental data. I am really missing a risk of bias assessment. The authors clearly state that it is not within the scope of their rapid review, but as it stands the review equates single centre retrospective studies with multicentre prospective observational studies. This review would be significantly strengthened by the addition of a risk of bias assessment. Results: 20 studies were excluded due to study design, but both observational and interventional studies were included. It should be made more clear as to why these studies were excluded. It seems as though the results that address the study question (are frailty measures valid in younger patients?) are buried in the supplemental data (Supplemental table S4 and Supplemental Figures 1a and 1b). I might recommend that this data be moved into the main body of the manuscript, and align the data with the study question (s). Thank you again for the opportunity to review this important work.
--	--

REVIEWER	Ken Sugimoto Kawasaki Medical University, Japan
REVIEW RETURNED	15-Dec-2020

GENERAL COMMENTS	The authors tried to identify how frailty is measured in younger populations and clarify if there is evidence showing patients' outcomes and care. According to their trial, limited evidence suggested that frailty measures based on the criteria for older populations had predictive validity in the younger populations, especially with specific diseases like HIV, CKD or ESRD, rheumatoid arthritis, or heart failure and receiving organ transplantation. The reviewer agrees with their findings because these diseases can accelerate patients' aging of organs. However, further investigations are needed to show the need for specific criteria of frailty for younger patients, as the authors mentioned. The studies designed only for younger subjects must be required
---

	to prove this issue because this review paper was nothing more than the one showing partial analysis of the studies designed for the subjects, including older adults. Since this paper would be the first review showing the current evidence of the significance of assessing frailty in younger populations, the reviewer recommends some modifications. Comments:  1. The reviewer recommends showing the figure showing the number of studies classified by diseases after the figure 2, and also recommends removing the supplementary table S2. The supplementary table S4 showed everything necessary. 2. The reviewer recommends putting the study number on the bar graph in the supplementary figure S1 and S2 to know which study shows the evidence of validity.
--	---

VERSION 1 – AUTHOR RESPONSE

	Comment	Response and revision
Reviewer 1	Introduction: 5 questions-where are the 5 answers? Some of the questions line up with section headers in the results section, but not all of them. Was there a hypothesis for this study?	We have amended the headings and sections throughout the findings section so that the findings align with the five questions. We hope this improves the overall findings section of the paper.
	Methods: Why was it not possible to involve the public? Engaging patients and and/or families might be incredibly instructive, especially when it comes to answering the 5th question (eg. what is the impact of frailty screening in younger patients). For clarity, I am not suggesting that engaging the public is a necessary revision for this manuscript, rather that it is insufficient to state that “it is not possible”.	This work was a rapid response to a request by the UK Department of Health and Social Care with a limited timeframe available for completion. Involving patients effectively in such rapid responsive work can be challenging, and in this case we did not feel that we could effectively engage with patients in the time available. Patient perspectives might have been most impactful in our analysis of question 5 (measuring impact). Patient involvement in rapid research in the pandemic has provided useful insights into how patients can be involved in rapid response work in the future (https://www.hra.nhs.uk/planning-and-improving-research/best-practice/public-involvement/public-involvement-pandemic-lessons-uk-covid-19-public-involvement-matching-service/) We have added the following sentence into the methods:

		This work was a rapid response to a request by the UK Department of Health and Social Care with a limited timeframe available for completion. Involving patients effectively in such rapid responsive work can be challenging, and in this case we did not feel that we could effectively engage with patients in the time available. However, the work was undertaken in response to a policy imperative.
	One of the key questions of the study is whether or not frailty measures are valid for patients under the age of 60. However, the rating scale (green, amber, red) isn't mentioned in the methodology, but appears in the supplemental data.	We have added our explanation of the colour coded framework to the synthesis section of the methods: Studies coded green demonstrated evidence of validity (a statistically significant prediction of mortality and/or hospital admissions, or authors described associations or agreement between measures as good or strong). Studies coded amber reported inconsistent or unclear evidence of validity, or authors described associations or agreement between measures as fair or modest. Studies coded red demonstrated no evidence of validity (the prediction of mortality and/or hospital admissions was not statistically significant, authors described associations or agreement between measures as weak or poor, or data were not reported in the publication to verify a claim of an association).
	I am really missing a risk of bias assessment. The authors clearly state that it is not within the scope of their rapid review, but as it stands the review equates single centre retrospective studies with multicentre prospective observational studies. This review would be significantly strengthened by the addition of a risk of bias assessment.	This point was raised by the editor and we have summarised our response to this on page 2 of this document.
	Results: 20 studies were excluded due to study design,	The 20 studies excluded due to study design were commentary

	but both observational and interventional studies were included. It should be made more clear as to why these studies were excluded.	pieces, literature reviews, or systematic reviews relating only to older populations. We have added this detail to the PRISMA flowchart.
	It seems as though the results that address the study question (are frailty measures valid in younger patients?) are buried in the supplemental data (Supplemental table S4 and Supplemental Figures 1a and 1b). I might recommend that this data be moved into the main body of the manuscript, and align the data with the study question (s).	We have been advised by Dr Sucksmith, senior assistant editor, to retain table S4 within the supplementary materials due to the table size. However, as suggested, we have moved figures 1a and b into the main document (now re-labelled Figures 3a and b).
Reviewer 2	1. The reviewer recommends showing the figure showing the number of studies classified by diseases after the figure 2, and also recommends removing the supplementary table S2. The supplementary table S4 showed everything necessary. Following further clarification: "Thank you for sending me this e-mail. I remember that my comment meant that the authors can create a new table showing the number of studies classified by diseases because the authors showed the table showing the detail of each study organized by author from A to Z.	As per your recommendation, we have removed table S2 from the supplementary materials. References to the table numbers in the main text, as well as subsequent table numbers, have been re-labelled accordingly. Thank you for also suggesting a table showing the number of studies classified by disease, and for your quick response to clarify the original comment. We have chosen not to add a further table showing the number of studies by diseases as this would duplicate the information in supplementary materials table S3 (formerly table S4), which provides detail of the study population.
	2. The reviewer recommends putting the study number on the bar graph in the supplementary figure S1 and S2 to know which study shows the evidence of validity.	We have been unable to add study numbers into the bar graph for figures S1 and S2. However, we have added a footnote to each figure to refer the reader to Table S3 (formerly S4, edited as per comment above), which details the studies these graphs refer to.

VERSION 2 – REVIEW

REVIEWER	Ken Sugimoto Kawasaki Medical University General Medical Center, Japan
REVIEW RETURNED	24-Feb-2021

GENERAL COMMENTS

The manuscript entitled "Measuring frailty in younger populations: a rapid review of evidence" was revised according to the reviewer's comment, and the responses were satisfied with the reviewer's demand.